# Comparative Outcomes of Intra-Aortic Balloon Pump Versus Percutaneous Left Ventricular Assist Device in High-Risk Percutaneous Coronary Intervention: A Systematic Review and Meta-Analysis

**DOI:** 10.3390/jcm14155430

**Published:** 2025-08-01

**Authors:** Dhiran Sivasubramanian, Virushnee Senthilkumar, Nithish Nanda Palanisamy, Rashi Bilgaiyan, Smrti Aravind, Sri Drishaal Kumar, Aishwarya Balasubramanian, Sathwik Sanil, Karthick Balasubramanian, Dharssini Kamaladasan, Hashwin Pilathodan, Kiruba Shankar

**Affiliations:** 1Department of Cardiology, Children’s Hospital of Philadelphia, Philadelphia, PA 19104, USA; 2Department of Gastroenterology, Mayo Clinic Hospital, Rochester, MN 55905, USA; virushnee25@gmail.com; 3Department of General Medicine, Coimbatore Medical College, Coimbatore 641018, India; nithishnanda127@gmail.com (N.N.P.); smrtiaravind@gmail.com (S.A.); baish2000@gmail.com (A.B.); dharssinik@gmail.com (D.K.); hashinpila@gmail.com (H.P.); 4Department of General Medicine, Dr. D. Y. Patil Medical College, Pune 411018, India; rashi.bilgaiyan@gmail.com; 5Department of Radiology, University of Pennsylvania, Philadelphia, PA 19104, USA; sridk@sas.upenn.edu; 6Department of Oncology, Sri Ramakrishna Hospital, Coimbatore 641018, India; sathwik98@gmail.com; 7Department of Critical Care Medicine, Christian Medical College, Vellore 632004, India; karthickbk76@gmail.com (K.B.); shankarkiruba04@gmail.com (K.S.)

**Keywords:** high-risk percutaneous coronary intervention, intra-aortic balloon pump, percutaneous left ventricular assist device, Impella, Tandem Heart, mechanical circulatory support

## Abstract

**Background/Objectives**: High-risk percutaneous coronary interventions (HR-PCIs) often require mechanical circulatory support (MCS) to maintain hemodynamic stability. Intra-aortic balloon pump (IABP) and percutaneous left ventricular assist device (PLVAD) are two commonly used MCS devices that differ in their mechanisms. We aimed to evaluate and compare the clinical outcomes associated with IABP and PLVAD use in HR-PCIs without cardiogenic shock. **Methods:** We conducted a search of PubMed, Scopus, Cochrane, Mendeley, Web of Science, and Embase to identify relevant randomized controlled trials and cohort studies, and we included 13 studies for the systematic review and meta-analysis. The primary goal was to define the difference in early mortality (in-hospital and 30-day mortality), major bleeding, and major adverse cardiovascular event (MACE) components (cardiogenic shock, acute kidney injury (AKI), and stroke/TIA) in IABP and PLVAD. We used a random-effects model with the Mantel–Haenszel statistical method to estimate odds ratios (ORs) and 95% confidence intervals. **Results:** Among 1 trial and 12 cohort studies (35,554 patients; 30,351 IABP and 5203 PLVAD), HR-PCI with IABP was associated with a higher risk of early mortality (OR = 1.53, 95% CI [1.21, 1.94]) and cardiogenic shock (OR = 2.56, 95% CI [1.98, 3.33]) when compared to PLVAD. No significant differences were found in the rates of arrhythmia, major bleeding, AKI, stroke/TIA, or hospital length of stay. **Conclusions**: In high-risk PCIs, PLVAD use is associated with lower early mortality and cardiogenic shock risk compared to IABP, with no significant differences in other major outcomes.

## 1. Introduction

High-risk percutaneous coronary interventions (HR-PCIs) are performed in patients who present with significant clinical challenges like advanced age, diabetes mellitus, multivessel disease, unprotected left main coronary artery involvement, complex coronary anatomy, and severely impaired left ventricular function, which increase their periprocedural risk. These procedures are complicated by transient interruptions of coronary blood flow caused by balloon inflation, contrast dye injections, guidewire passage, and stent manipulation, which can result in the acute impairment of cardiac function [1,2]. Mechanical circulatory support (MCS) devices, such as percutaneous left ventricular assist devices (PLVADs) and intra-aortic balloon pump (IABP), are employed to enhance hemodynamic stability by ensuring adequate systemic and coronary perfusion during these procedures [2,3].

Intra-aortic balloon pump (IABP) has been a mainstay device in providing such support; however, in recent times, IABP’s limited augmentation of cardiac output has promoted the growth of the use of PLVAD, which includes Impella and Tandem Heart [1,4].

Several studies and a randomized controlled trial (RCT) have compared IABP and PLVAD use in HR-PCI, but the findings remain inconsistent. Previous meta-analyses comparing IABP and PLVAD included both HR-PCIs and shock populations, which may confound the interpretation of outcomes specific to elective HR-PCIs [4,5,6]. Although there is only one RCT to date [7], several cohort studies have emerged in recent years. In this systematic review and meta-analysis, we aimed to strictly compare the outcomes of IABP and PLVAD in HR-PCIs without cardiogenic shock to provide more targeted clinical insights.

## 2. Methods

### 2.1. Search Strategy

We conducted a systematic search of the PubMed, Scopus, Cochrane, Mendeley, Web of Science, and Embase databases to identify relevant RCTs and cohort studies with patients undergoing HR-PCI under MCS with IABP or PLVAD up to June 2025. The search was limited to English-language articles. The following search terms were included in various combinations: “intra-aortic balloon pump”, “percutaneous left ventricular assist device”, “Impella”, “Tandem Heart”, “high-risk percutaneous coronary intervention”, “protected percutaneous coronary intervention”, “randomized”, and “studies”. The search strategy was performed with adherence to the Preferred Reporting Items for Systematic Reviews and Meta-Analysis (PRISMA) guidelines (Figure 1) [8,9]. Studies using the same patient databases were carefully excluded to prevent data overlap. Two independent authors (DS and VS) reviewed the search results separately to select the studies based on inclusion and exclusion criteria. Reference lists of the included studies were reviewed for any missing relevant studies. Disagreements were resolved by a third and fourth reviewer (NNP and RB). This review was not registered in any registry.

### 2.2. Inclusion Criteria

The inclusion criteria were cohort studies or RCTs with at least 10 patients undergoing HR-PCI with hemodynamic support that compared the differences based on whether the patients had PLVAD or IABP and reported one or all of the following outcomes: early mortality (in-hospital and 30-day mortality) and major cardiovascular adverse event (MACE) components such as cardiogenic shock, major bleeding, acute kidney injury (AKI), arrhythmia, and stroke/transient ischemic attack (TIA). Only adult patients >18 years old were included. Abstracts from major journals were included if they provided the desired results.

### 2.3. Exclusion Criteria

Non-English literature, systematic reviews, meta-analysis, narrative reviews, case reports/series, editorials, study protocols, commentaries, editorial letters, and studies that report the outcomes for only one intervention without the other were excluded. Studies with patients aged < 18 years were also excluded.

### 2.4. Data Extraction and Quality Assessment

Six independent investigators were tasked with data extraction from the included studies, and a thorough review of the data was performed. For each study, the data on the number of events and the total number were abstracted.

The extracted data included primary outcomes of interest, early mortality (including 30-day mortality and in-hospital mortality), peri- and post-procedural cardiogenic shock, major bleeding events, new arrhythmias, the occurrence of AKI and stroke/TIA, along with mean length of stay in the hospital.

Using the Newcastle–Ottawa scale (NOS) risk-of-bias assessment instrument, a total of 13 studies from the initial 155 records that met the inclusion criteria were assessed. After two impartial assessors evaluated the potential for bias, a final table was created based on their consensus (Table 1 and Table 2).

### 2.5. Statistical Analysis

The Cochrane Collaboration and the Meta-analysis of Observational Studies in Epidemiology (MOOSE) guidelines were fully adhered to in this meta-analysis [22]. We used the Cochrane Collaboration’s “Review Manager” software version 5.4.1 (The Cochrane Collaboration, London, UK) for data analysis. We employed the Mantel–Haenszel random-effects model to estimate the odds ratios (ORs) and the associated 95% confidence intervals (CIs) for binary outcomes. The inverse variance method was also utilized to determine the weighted mean difference (MD) for continuous outcomes.

Forest plots were created for the graphical representation of the clinical outcomes. Heterogeneity between studies, defined as variation among the results of individual studies beyond that expected from chance, was evaluated with the I^2^ statistic. Interpretation for I^2^: 50%, low heterogeneity; 50–75%, moderate heterogeneity; and >75%, high heterogeneity. Publication bias was evaluated through visual inspection of funnel plots for asymmetry in Review Manager.

## 3. Results

A total of 13 studies (1 RCT and 12 cohort studies) met our criteria and were included in this study from a total of 155 records that were initially identified from the databases (Figure 1). A total of 35,554 patients were included in this analysis conducted between 2014 and 2025, with 30,351 patients receiving IABP and 5203 receiving PLVAD. The study characteristics are presented in Table 3. The study by Al-Khadra et al. [23] using the National Inpatient Sampling (NIS) database met the inclusion criteria but was carefully excluded because there was a potential overlap of patients as they used the same database as Khera et al. [14].

### 3.1. Mortality

Early mortality (in-hospital and 30-day mortality) was significantly higher in the IABP group when compared to the PLVAD group (OR = 1.53, 95% CI [1.21, 1.94], *p* < 0.001, I^2^ = 50%) (Figure 2).

### 3.2. Subgroup Analysis

We observed a significantly higher incidence of cardiogenic shock in the IABP group compared to the PLVAD group (OR = 2.56, 95% CI [1.98, 3.33], *p* < 0.00001, I^2^ = 69%) (Figure 3). We also observed no significant difference in major bleeding (OR = 0.64, 95% CI [0.33, 1.25], *p* = 0.19, I^2^ = 75%) (Figure 4), stroke/TIA (OR = 1.23, 95% CI [0.90, 1.69], *p* = 0.20, I^2^ = 73%) (Figure 5), arrhythmia (OR = 0.72, 95% CI [0.26, 1.97], *p* = 0.52, I^2^ = 73%) (Figure 6), AKI (OR = 1.01, 95% CI [0.87, 1.16], *p* = 0.90, I^2^ = 0%) (Figure 7), or length of stay (MD = −1.04, 95% CI [−3.72, 1.65], *p* = 0.45, I^2^ = 0%) (Figure 8) between both of the groups.

### 3.3. Heterogeneity

Early mortality demonstrated low heterogeneity across the studies (I^2^ = 50%), indicating relatively consistent findings. The cardiogenic shock, major bleeding, arrhythmia, and stroke/TIA outcomes showed moderate heterogeneity (I^2^ = 69%, 75%, 73%, and 73%, respectively), suggesting some variability in the effect estimates. AKI and length of hospital stay had no heterogeneity (I^2^ = 0%), reflecting highly consistent results among the studies.

### 3.4. Assessment of Publication Bias

The funnel plot (Figure 9) evaluating early mortality suggests some degree of publication bias, evidenced by the asymmetry. Larger studies appear more consistent and tend to cluster at OR = 1–2, suggesting less extreme estimates and higher reliability, while smaller studies vary widely, which is expected due to higher random error in smaller studies.

The funnel plot (Figure 10) evaluating cardiogenic shock suggests high heterogeneity, as seen from the wide range of ORs (from <0.1 to >10). The asymmetry also suggests the potential for publication bias, especially against small studies showing a benefit (OR < 1).

## 4. Discussion

During HR-PCI, patients are subject to increased risk of hemodynamic collapse; to mitigate these risks, MCS devices are used to maintain adequate systemic and coronary perfusion during the procedure [1].

As of now, there are no clear criteria to identify high-risk PCI candidates. The Society for Cardiovascular Angiography & Interventions (SCAI) specifically highlights that those patients with severely impaired ejection fraction (≤35%), unprotected left main coronary artery, or complex multivessel disease undergoing PCI are considered to be at high risk of hemodynamic compromise and may benefit from MCS. The SCAI also notes that additional factors, such as comorbidities, advanced age, anticipated procedural complexity (e.g., atherectomy), and extent of myocardium at risk, should be integrated into the risk assessment for MCS use [24].

The commonly used MCS devices in HR-PCI are IABP and PLVAD. PLVAD comprises two main devices, Impella 2.5 and Tandem Heart [2]. Both groups of devices have distinct mechanisms of action, and a comparison of their effectiveness in HR-PCI is much needed.

### 4.1. Outcomes

#### 4.1.1. Early Mortality

Our analysis reported lower early mortality with PLVAD when compared to IABP. This is in alignment with the findings of a large propensity-adjusted analysis conducted by Lansky et al. [15] of 2156 patients undergoing HR-PCI, which showed that Impella was associated with improved in-hospital survival, lower myocardial infarction, and cardiogenic shock events compared to IABP, while other MACEs were similar in both groups. The only RCT that compared Impella and IABP in HR-PCI is the PROTECT II trial [7], which enrolled 448 patients based on anatomical and clinical criteria, and they showed no significant difference in 30-day mortality between the groups. However, Impella showed a potential benefit over IABP in terms of 90-day mortality. Additionally, there was a non-significant trend towards fewer MACEs with Impella.

#### 4.1.2. Cardiogenic Shock

A significant increase in the incidence of cardiogenic shock was noted in our study while using IABP compared to PLVAD. This can perhaps be attributed to the difference in the mechanism of the two devices. The IABP works by inflating during diastole to augment coronary perfusion and deflating just before systole to reduce afterload and thereby moderately increasing the cardiac output and improving coronary perfusion. Impella actively unloads the left ventricle (LV) by aspirating blood from the LV and expelling it into the aorta, while Tandem Heart withdraws blood from the left atrium and returns it to the femoral artery, bypassing the LV, thereby increasing the cardiac output and lowering the myocardial oxygen demand [25].

#### 4.1.3. Stroke

Our analysis showed that the occurrence of stroke was similar in the Impella and PLVAD groups. Stroke incidence is generally higher with PLVAD, including Impella and Tandem Heart, compared to IABP in high-risk percutaneous procedures, according to previous studies. In cardiogenic shock, stroke rates range from 4 to 8% with PLVAD versus 3 to 4% with IABP [26,27,28]. In HR-PCI without shock, the absolute stroke rates are low, with no significant difference between the devices [15,29]. These trends are consistent across large observational studies and meta-analyses [15,26,29].

#### 4.1.4. Major Bleeding

Our analysis showed that the occurrence of major bleeding was similar in the Impella and PLVAD groups. According to previous studies, major bleeding is significantly more common with PLVAD than with IABP in HR-PCI. Other meta-analyses report a 2- to 3-fold increased risk with PLVAD (RR 2.35–2.85) [26,30]. During MCS use in cardiogenic shock, the in-hospital major bleeding rates are 31.3% with PLVAD versus 16.0% with IABP [31], with similar trends in real-world data showing higher hematologic and overall complications with PLVADs [26,28]. The bleeding rates generally range from 11 to 19% for IABP and 19 to 31% for PLVAD in HR-PCI or shock [26,32,33,34].

#### 4.1.5. Acute Kidney Injury

AKI is a major risk after high-risk PCI, especially in patients with low LVEF or chronic kidney disease (CKD). Our study does not show a significant difference in the occurrence of AKI between the two groups. This, however, is in contrast to Flaherty MP et al., who found that the use of PLVAD (Impella) significantly reduces AKI risk and lowers dialysis rates, likely by improving renal perfusion [35]. In contrast, IABP use increases AKI risk, likely due to insufficient hemodynamic support [31,36].

#### 4.1.6. Arrhythmia

We found no significant difference in the incidence of arrhythmia between the two groups. A previous meta-analysis pooled from RCTs comparing IABP and PLVADs in high-risk PCI showed similar results [29]; these findings are consistent across all of the available randomized data and observational studies, such as the large propensity-weighted retrospective analysis of 2879 patients by Miller et al. [21].

#### 4.1.7. Length of Stay and Cost Effectiveness

The length of hospital stay is similar between the two groups. Multicenter retrospective and prospective studies have not shown any significant differences [10,12,13,18].

PLVADs are significantly less cost-effective than IABP in this setting. Contemporary claims-based analyses by Miller et al. showed that the mean total costs for the index admission are substantially higher with PLVADs (e.g., Impella) compared to IABP (approximately USD 96,700 vs. USD 71,900; *p* < 0.001), and this cost difference persists at 30 days post-procedure [21].

### 4.2. Strengths and Limitations

Our study results are different from previous meta-analyses [4,5,6]; the exclusive focus on high-risk PCI without cardiogenic shock addresses a critical gap. Previous meta-analyses have pooled these distinct populations, despite their pathophysiology, risk profiles, and expected outcomes. By deliberately excluding studies involving cardiogenic shock, the current work provides a more precise estimate of the comparative effectiveness and safety of IABP vs. PLVAD specifically in high-risk PCI settings, where the risk–benefit calculus may differ substantially from that in shock. It offers a focused, up-to-date synthesis of evidence for device selection in high-risk PCI, unconfounded by the inclusion of cardiogenic shock populations. This meta-analysis further incorporates the more recent registry data up to 2025, which enhances the relevance of the findings compared to earlier meta-analyses that relied on older or less comprehensive datasets. We showed a significantly higher early mortality and cardiogenic shock in IABP compared to PLVAD. We also included the largest sample size of 35,554 patients compared to other analyses. Although one of the meta-analyses that reported PLVAD could be associated with a higher occurrence of adverse events such as bleeding and AKI [6], we did not find any significant difference in MACEs other than cardiogenic shock between the groups.

This systematic review and meta-analysis has several limitations.

First, there was significant heterogeneity among the included studies, particularly in secondary outcomes such as major bleeding, stroke/TIA, cardiogenic shock, and arrhythmia, which may reflect differences in the study design, including randomized controlled trials versus observational studies, sample size, and follow-up duration, resulting in variable risk of bias and event ascertainment, directly impacting the outcome estimates and their variability across studies.

The patient selection criteria are inconsistent, with studies enrolling populations that differ in age, comorbidities, baseline left ventricular function, and procedural risk, all of which influence complication rates. For example, some studies include patients with more advanced coronary disease or higher bleeding risk, leading to divergent rates of major bleeding and arrhythmia [26,29].

Procedural techniques and operator skill also vary widely. Device selection (e.g., Impella vs. Tandem Heart), the vascular access approach, anticoagulation protocols, and periprocedural management are not standardized, and operator and center experience with newer devices can affect complication rates, particularly for bleeding and vascular events [6,29,37].

Definitions of adverse events such as major bleeding, stroke, and arrhythmia are not uniform across studies, with some using clinical criteria and others relying on administrative or registry data, further increasing heterogeneity [29,37]. Furthermore, the funnel plots show asymmetry towards IABP, suggesting that studies favoring PLVAD or higher mortality or cardiogenic shock in IABP may be more likely to be published.

Second, the included studies also varied in their reporting of mortality outcomes; some reported in-hospital mortality, while others reported 30-day mortality. To account for this inconsistency, we grouped these endpoints under the broader category of early mortality, which may obscure important differences. In-hospital mortality is more likely to reflect procedural and immediate post-procedural complications, whereas 30-day mortality includes later events such as subacute heart failure or non-cardiac causes. Prior studies have shown that the risk of death is highest in the first week following PCI and declines thereafter, with cardiac causes predominating early and non-cardiac causes increasing over time [38,39]. As a result, combining these endpoints may dilute the impact of device-related procedural complications on the observed outcomes.

Third, the majority of included studies were observational cohort studies rather than randomized controlled trials. As a result, the findings may be subject to inherent biases, including confounding by indication. For example, clinicians may preferentially select PLVADs, particularly Impella, for patients perceived to be at lower risk or with specific anatomical considerations, while IABP is reserved for sicker patients due to easier insertion and familiarity. This selection bias can skew outcome comparisons. While few observational studies have statistical adjustments, such as propensity score matching and multivariate regression to address baseline imbalances [15,21,23], residual confounding from unmeasured or unknown variables cannot be excluded. The analysis by Agoritas et al. explicitly notes that even with advanced adjustment methods, non-randomized studies are inherently susceptible to selection and referral bias, and only randomization can fully balance both known and unknown confounders. This limitation should temper the interpretation of any observed associations, particularly regarding device selection and outcomes [40].

Fourth, the studies do not specify the exact distinction between intra-procedural and post-procedural cardiogenic shock, so we could not analyze these separately. Additionally, there are no large-scale studies or randomized controlled trials that specifically differentiate between intra-procedural and post-procedural cardiogenic shock in patients undergoing HR-PCI when comparing IABP and PLVAD. The guidelines from the American College of Cardiology, American Heart Association, and Society for Cardiovascular Angiography & Interventions also do not address this distinction, reflecting the lack of granular data in the literature [41,42]. This limitation affects the interpretation of device efficacy and safety, as intra-procedural shock may be more directly related to procedural complications and device performance, whereas post-procedural shock may reflect broader patient comorbidities or delayed complications. Without this distinction, attributing outcomes to device choice versus patient or procedural factors remains challenging.

Fifth, the lack of granular data regarding device type and procedural details limited our ability to perform device-specific subgroup analyses. While most PLVAD usage likely involved Impella 2.5, the exact distribution between Impella 2.5, Impella 5.0, and Tandem Heart devices were not uniformly reported. The Tandem Heart requires transseptal puncture, increasing procedural complexity and risk, but offers greater support than the Impella 2.5. The Impella 5.0 is comparable to Tandem Heart in terms of support. Differences in the flow capacity, mechanism, complication profiles, and access requirements between these devices could impact the outcomes, and these could not be analyzed separately in our study. This limitation is particularly relevant because prior analyses have shown that Impella and Tandem Heart may have different rates of major bleeding and short-term mortality in the HR-PCI population. Therefore, the results should be interpreted with caution, and the generalizability of the findings to specific devices or clinical scenarios is limited [6,29,43].

Lastly, the protocol was not prospectively registered in any registry (e.g., PROSPERO), which may introduce concerns regarding transparency and the potential for selective reporting.

### 4.3. Future Directions

It is important to note that Impella 2.5 is the most used form of commercially available PLVAD, which has a flow rate of 2.5 L/min, which was the device used in the PROTECT II trial. In contrast, Impella 5 provides more robust support with a flow rate of 5 L/min but requires a surgical cutdown of the femoral or axillary artery, similar to Tandem Heart, which limits its practicality in the clinical setting [41]. A study by Kovacic et al. [25] compared Tandem Heart and Impella 2.5 in 68 patients who underwent HR-PCI using either device, and they found that there was no difference in the short- or long-term clinical outcomes between the two groups. Large-scale multicenter RCTs are needed to clarify whether there is a difference between Impella 2.5 and Impella 5 or between these and Tandem Heart.

Additionally, the findings from the PRAISE registry show that the 1-year outcomes between STEMI (ST-elevated myocardial infarction) and NSTEMI (non-ST-elevated myocardial infarction) are comparable once adjusted for comorbidities and revascularization completeness [44]. This underscores the need to individualize MCS decisions based on patient profiles rather than MI type alone.

Our analysis exclusively focuses on short-term outcomes, such as in-hospital and 30-day mortality. While these endpoints provide valuable insights into early procedural risk, they do not capture the full clinical trajectory following HR-PCI. Future studies should incorporate long-term outcomes such as 6- or 12-month mortality and rehospitalization to better assess the sustained effectiveness of PLVADs over IABP in HR- PCI. Current evidence is limited to short-term endpoints, and standardized long-term follow-up is needed to guide clinical decision-making.

## 5. Conclusions

PLVAD use in HR-PCI was associated with lower early mortality and cardiogenic shock compared to IABP, with no significant difference in other adverse events and length of stay. While these findings suggest a potential clinical benefit of PLVADs, this must be weighed against their significantly higher cost. Moreover, as these results are derived predominantly from observational data, they should be interpreted with caution due to the potential for confounding by indication. Further, randomized controlled trials are needed to confirm these results and guide device selection.

## Figures and Tables

**Figure 1 jcm-14-05430-f001:**
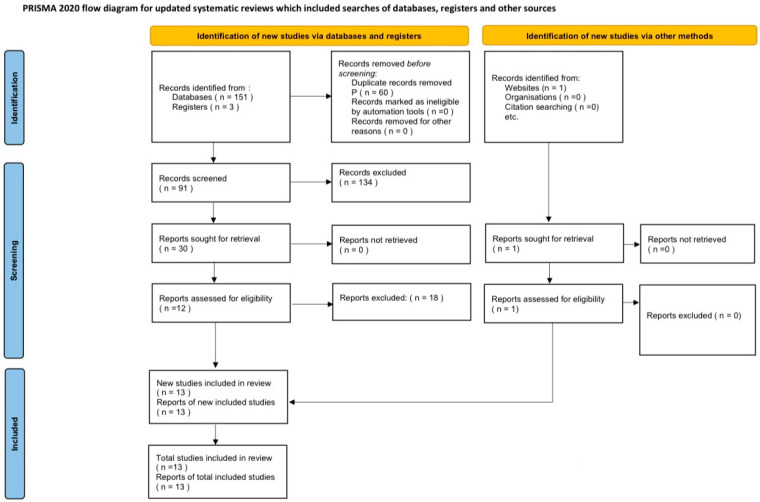
Flowchart of study selection process using PRISMA guidelines [8,9].

**Figure 2 jcm-14-05430-f002:**
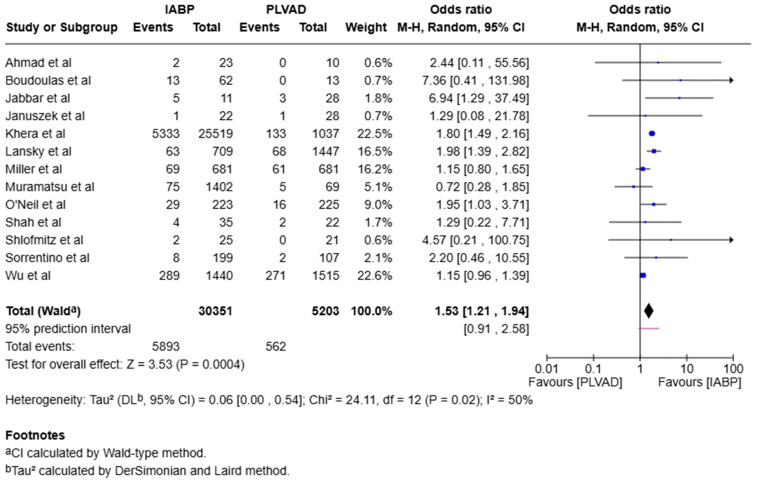
Forest plot showing the odds ratio (OR) of early mortality (in-hospital and 30-day mortality) in the IABP and PLVAD groups. IABP: intra-aortic balloon pump; PLVAD: percutaneous left ventricular assist device [7,10,11,12,13,14,15,16,17,18,19,20,21].

**Figure 3 jcm-14-05430-f003:**
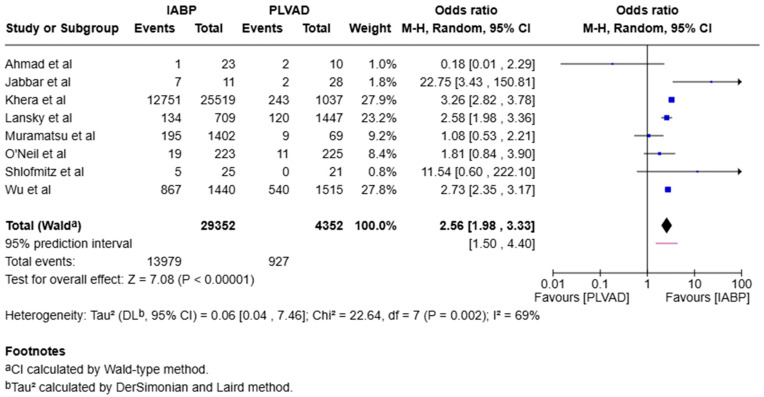
Forest plot showing the odds ratio (OR) of cardiogenic shock in the IABP and PLVAD groups. IABP: intra-aortic balloon pump; PLVAD: percutaneous left ventricular assist device [7,10,12,14,15,16,18,20].

**Figure 4 jcm-14-05430-f004:**
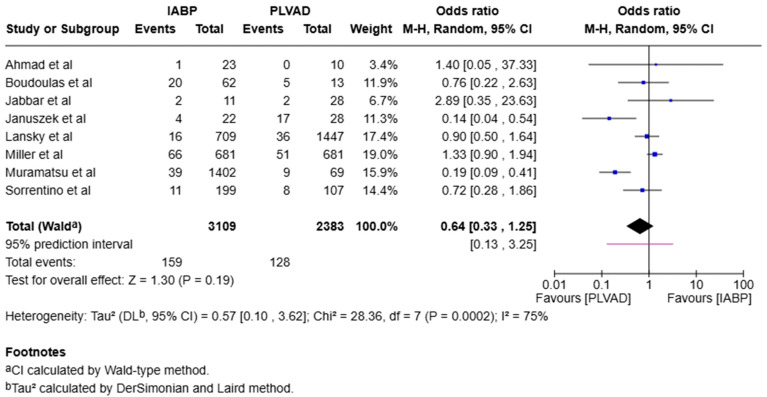
Forest plot showing the odds ratio (OR) of major bleeding in the IABP and PLVAD groups. IABP: intra-aortic balloon pump; PLVAD: percutaneous left ventricular assist device [10,11,12,13,15,16,19,21].

**Figure 5 jcm-14-05430-f005:**
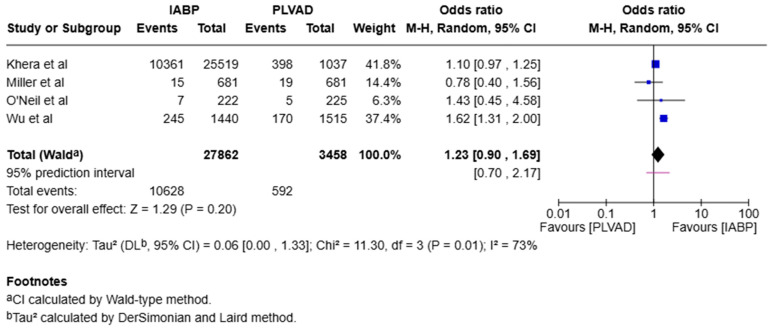
Forest plot showing the odds ratio (OR) of stroke/TIA in the IABP and PLVAD groups. IABP: intra-aortic balloon pump; PLVAD: percutaneous left ventricular assist device [7,14,20,21].

**Figure 6 jcm-14-05430-f006:**
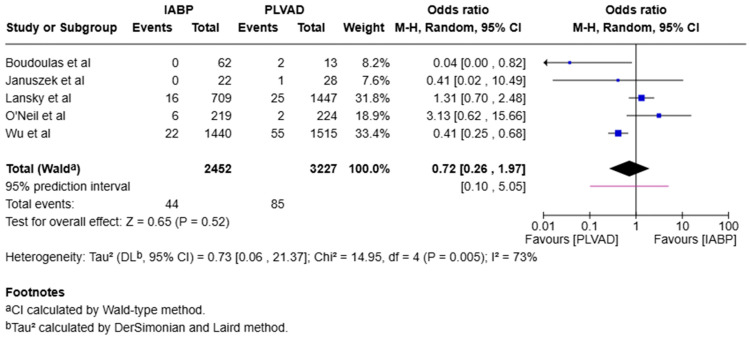
Forest plot showing the odds ratio (OR) of arrhythmia in the IABP and PLVAD groups. IABP: intra-aortic balloon pump; PLVAD: percutaneous left ventricular assist device [7,11,13,15,20].

**Figure 7 jcm-14-05430-f007:**
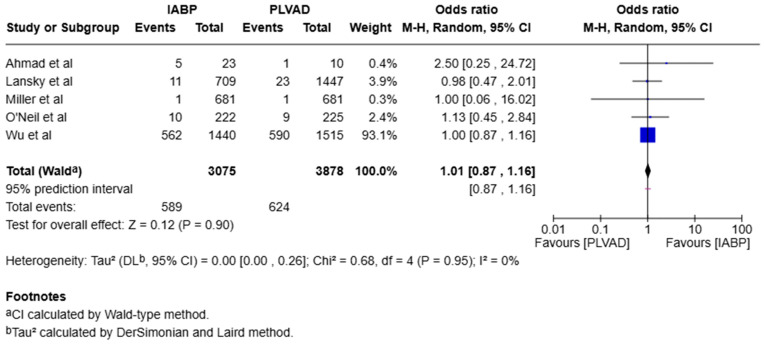
Forest plot showing the odds ratio (OR) of acute kidney injury in the IABP and PLVAD. IABP: intra-aortic balloon pump; PLVAD: percutaneous left ventricular assist device groups [7,10,15,20,21].

**Figure 8 jcm-14-05430-f008:**
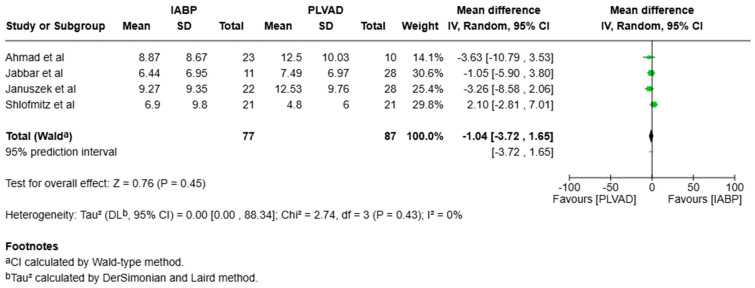
Forest plot showing the mean difference (MD) in length of stay in the IABP and PLVAD groups. IABP: intra-aortic balloon pump; PLVAD: percutaneous left ventricular assist device [10,12,13,18].

**Figure 9 jcm-14-05430-f009:**
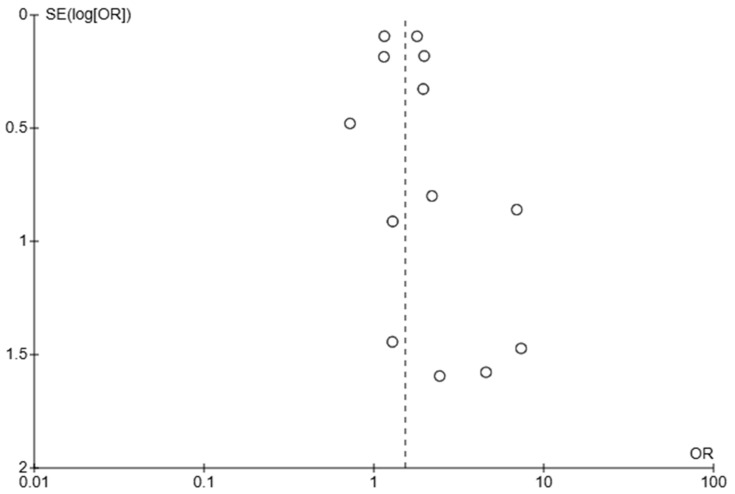
Funnel plot using odds ratio (OR) data for early mortality (in-hospital and 30-day mortality).

**Figure 10 jcm-14-05430-f010:**
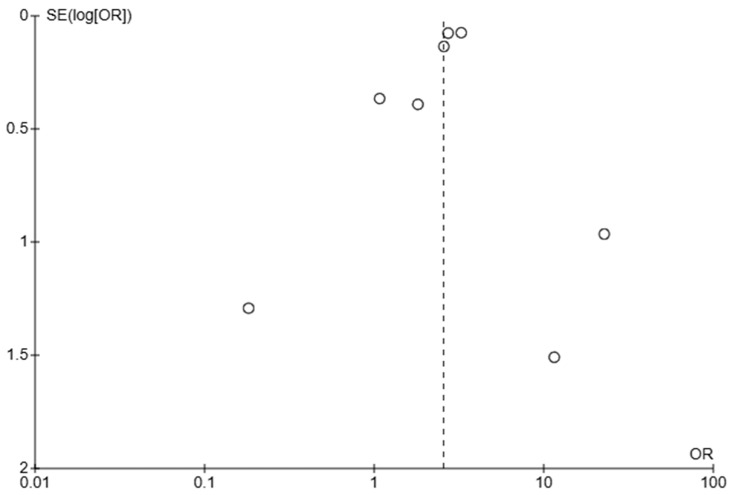
Funnel plot using odds ratio (OR) data for cardiogenic shock.

**Table 1 jcm-14-05430-t001:** Risk-of-bias assessment of cohort studies using NOS. NOS: Newcastle–Ottawa scale. *: present, 0: not present.

Studies	Selection			Comparability			Outcome			
	Representativeness of the Exposed	Selection of the Non-Exposed	Ascertainment of Exposure	Outcome of Interest Not Present at Study Beginning	Main Factor	Additional Factor	Assessment	Follow-Up Length	Adequacy of Follow-Up	Score
Ahmed et al. [10]	*	*	*	*	*	*	*	*	*	9
Boudoulas et al. [11]	*	*	*	*	0	0	*	*	0	6
Jabbar et al. [12]	*	*	*	*	0	0	*	*	0	6
Januszek et al. [13]	*	*	*	*	0	0	*	*	*	7
Khera et al. [14]	*	*	*	*	*	*	*	*	*	9
Lansky et al. [15]	*	*	*	*	*	*	*	*	*	9
Muramatsu et al. [16]	*	*	*	*	0	0	*	*	*	9
Shah et al. [17]	*	*	*	*	*	0	*	*	*	8
Shlofmitz et al. [18]	*	*	*	*	*	*	*	*	*	9
Sorrentino et al. [19]	*	*	*	*	*	*	*	*	0	8
Wu et al. [20]	*	*	*	*	*	*	*	*	*	9
Miller et al. [21]	*	*	*	*	*	*	*	*	*	9

**Table 2 jcm-14-05430-t002:** Risk-of-bias assessment of PROTECT II trial [7] using Cochrane risk-of-bias (ROB2) tool.

Risk-of-Bias Assessment—Protect II Trial (O’Neill et al. [7])		
Domain	Judgment	Justification
Random sequence generation (selection bias)	Low risk of bias	Randomization was performed using an automated interactive voice response system stratified by geography and angioplasty indication, indicating an adequate randomization process.
Allocation concealment (selection bias)	Unclear risk of bias	While randomization was automated, the method of allocation concealment (e.g., central allocation, opaque envelopes) was not explicitly stated in the methods section.
Blinding of participants and personnel (performance bias)	High risk of bias	This was an open-label trial with no blinding of clinicians or participants. Operator awareness may have influenced treatment decisions, such as more aggressive atherectomy in the Impella arm.
Blinding of outcome assessment (detection bias)	Low risk of bias	Outcome events were adjudicated by an independent Clinical Events Committee (CEC) blinded to treatment assignment, minimizing detection bias.
Incomplete outcome data (attrition bias)	Low risk of bias	Very low dropout rate reported: 448 out of 452 randomized patients were included in the final analysis. Both ITT and PP analyses were clearly defined.
Selective reporting (reporting bias)	Unclear risk of bias	Although the primary and secondary outcomes were pre-specified, the study was stopped early due to futility, and additional subgroup analyses were emphasized, raising the possibility of selective emphasis in reporting.
Other bias	High risk of bias	Potential learning curve effect was acknowledged, with safety improvements noted later in the trial. This introduces a possible performance or temporal bias in early vs. late enrollees.

**Table 3 jcm-14-05430-t003:** Main characteristics of the selected studies. PLVAD: percutaneous left ventricular assist device; IABP: intra-aortic balloon pump.

Study ID	Study Center	Study Duration (Year–Year)	Study Design	PLVAD	Number of Patients		AGE	AGE
					IABP	PLVAD	IABP	PLVAD
Sorrentino et al. (2017) [19]	Single-center prospective study	2007 to 2016	Prospective cohort	Impella 2.5	199	107	70 ± 10.9	70 ± 10.9
Ahmed et al. (2015) [10]	Observational single-center study	2008 to 2014	Retrospective cohort	Impella 2.5	23	10	71 ± 14.47	71 ± 14.47
O’Neil et al. (2015) [7]	Prospective multicenter randomized trial	2007 to 2010	Randomized controlled trial	Impella 2.5	223	225	67 ± 11	68 ± 11
Khera et al. (2015) [14]	NIS database	2004 to 2012	Retrospective cohort	Not specified	25519	1037	64.7	69
Januszek et al. (2022) [13]	Single-center prospective study	2018 to 2021	Prospective cohort	Impella 2.5	22	28	74.6 ± 9.6	7.6 ± 9.6
Boudoulas et al. (2012) [11]	Observational single-center study	2008 to 2010	Retrospective analysis	Impella 2.5	62	13	60.8 ± 12.6	62.5 ± 9.7
Shah et al. (2012) [17]	Observational single-center study	2007 to 2009	Prospective analysis	Tandem Heart 35 and Impella	35	22	60 ± 9.9	69 ± 9
Jabbar et al. (2022) [12]	Observational single-center study	2010 to 2014	retrospective cohort	Impella 2.5	11	28	73.73 ± 16.89	75.60 ± 10.78
Muramatsu et al. (2022) [16]	Japanese percutaneous coronary intervention registry	2018	Prospective cohort	Impella 2.5	1402	69	74	74
Lansky et al. (2022) [15]	Premier Healthcare Database	2016–2019	Retrospective cohort	Impella 2.5	709	1447	69.2 ± 10.9	71.4 ± 10.9
Wu et al. (2020) [20]	NIS database	2016	retrospective cohort	Impella 2.5	1440	1515	NA	NA
Shlofmitz et al. (2017) [18]	Observational multicenter study	2011–2017	Retrospective cohort	Impella 2.5	25	21	NA	NA
Miller et al. (2025) [21]	Observational multicenter study	2016–2022	Retrospective cohort	Impella 2.5	681	681	65 ± 12.3	71 ± 11.8

## Data Availability

All data generated or analyzed during this study are included in this published article, available from the corresponding author (drdhiransivasubramanian@gmail.com) upon reasonable request.

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
