# Peer review of "Comparative Outcomes of Intra-Aortic Balloon Pump Versus Percutaneous Left Ventricular Assist Device in High-Risk Percutaneous Coronary Intervention: A Systematic Review and Meta-Analysis"

_jcm, 2025, doi:10.3390/jcm14155430_

Round 1
Reviewer 1 Report
Comments and Suggestions for Authors
Thank you for asking me to review this meta-analysis that compares IABP vs pVLAD for high risk PCI support.
This is a relevant study which tries to answer a very pertinent question that is of interest to clinicians in the field. The abstract is balanced and well describes the study and its findings. The aim of the study is well documented in Introduction. The studies included were from 2015 to 2022 and therefore is a little dated now. The manuscript is well written and confirms to the PRISMA and MOOSE guidelines for reporting studies and meta-analysis. A number of databases were searched with careful identification of appropriate studies. Search criteria was adequate for identification of the studies and bias assessment was adequately performed by 6 reviewers.
The methodology is well described and analysis is well conducted using standard software. Statistical methods ae well used and described. The results are clearly presented with very good illustrations and figures. Conclusions are consistent with the presented results. Limitations have been well documented. The Bibliography is adequate and up to date for the period of the study but some more recent articles could be included with appropriate additions in the discussion section. In the discussion, please talk about and include some recent references with cost effectiveness analysis of the two devices.
Author Response
Reviewer 1#
Comment 1: The Bibliography is adequate and up to date for the period of the study but some more recent articles could be included with appropriate additions in the discussion section.
Response 1: Thank you for this suggestion. We have included the most recent large-scale retrospective study by Miller et al in 2025 on IABP vs PLVAD in HR-PCI without cardiogenic shock in our review and meta-analysis data. We have also further added a few more articles and elaborated our discussion on the limitations and strengths of our study.
Page no.7 in the table, Page no.13 in the discussion
Comment 2: In the discussion, please talk about and include some recent references with cost-effectiveness analysis of the two devices.
Response 2: We have included the cost-effectiveness subheading and discussed it further in detail in the discussion session, citing recent articles. PLVAD is less cost-effective than IABP in HR-PCI according to the recent registry data and studies.
Page no. 13 under discussion and subheading 4.1.7 Length of Stay & Cost effectiveness

Reviewer 2 Report
Comments and Suggestions for Authors
Congratulations to the authors for their optimal manuscript; here my comments;
The authors should clearly delineate how this meta-analysis advances beyond prior publications. For example, highlight the inclusion of more recent registry data, the exclusive focus on high-risk percutaneous coronary interventions , and the deliberate exclusion of studies involving cardiogenic shock. This distinction will clarify the unique contribution of the current work.
Several of the reported outcomes demonstrate moderate to high heterogeneity—notably, major bleeding (I²=66%), stroke or transient ischemic attack (I²=73%), and arrhythmia (I²=79%). These findings are mentioned but not thoroughly addressed in the discussion. The authors are encouraged to expand on this point, exploring possible sources of heterogeneity such as variations in study design, patient selection, or procedural techniques.
Additionally, the analysis combines all percutaneous left ventricular assist devices, including Impella 2.5, Impella CP, and TandemHeart—into a single pooled group, despite differences in flow capacity and hemodynamic support. If granular data are available, the authors should conduct separate subgroup analyses for Impella and TandemHeart. If this is not feasible, it is recommended to explicitly include this limitation in the discussion and explain how device heterogeneity may affect the results. Moreover the authors are encouraged to include the latest evidences of different prognosis in ACS patients in their discussion (https://doi.org/10.1007/s40256-025-00739-8)
Finally, some forest plots are suboptimally formatted, lacking detailed axis labels, consistent confidence interval shading, and clear risk tables. The authors should improve this aspect by using uniform labeling, adding 95% confidence interval bars, and including risk tables beneath the plots.
Author Response
Reviewer 2#
Comment 1: The authors should clearly delineate how this meta-analysis advances beyond prior publications. For example, highlight the inclusion of more recent registry data, the exclusive focus on high-risk percutaneous coronary interventions, and the deliberate exclusion of studies involving cardiogenic shock. This distinction will clarify the unique contribution of the current work.
Response 1: Thank you for this suggestion. We have highlighted the exclusive focus on HR-PCI without cardiogenic shock in detail in our revised manuscript under discussion, clarifying the unique contribution of this work without confounding through the inclusion of the cardiogenic shock population.
Page 13 under subheading 4.2 Strengths & Limitations
Comment 2: Several of the reported outcomes demonstrate moderate to high heterogeneity—notably, major bleeding (I²=66%), stroke or transient ischemic attack (I²=73%), and arrhythmia (I²=79%). These findings are mentioned but not thoroughly addressed in the discussion. The authors are encouraged to expand on this point, exploring possible sources of heterogeneity such as variations in study design, patient selection, or procedural techniques.
Response 2: We have now described in detail the heterogeneity of the studies and also expanded on possible sources of heterogeneity, such as operator skill, study design, device selection, and patient selection criteria, in the discussion section.
Page 14 under 4.2 Strength & limitations section.
Comment 3: Additionally, the analysis combines all percutaneous left ventricular assist devices, including Impella 2.5, Impella CP, and TandemHeart—into a single pooled group, despite differences in flow capacity and hemodynamic support. If granular data are available, the authors should conduct separate subgroup analyses for Impella and TandemHeart. If this is not feasible, it is recommended to explicitly include this limitation in the discussion and explain how device heterogeneity may affect the results. Moreover the authors are encouraged to include the latest evidences of different prognosis in ACS patients in their discussion (https://doi.org/10.1007/s40256-025-00739-8)
Response 3: We had already described the limitation of our study based on the unavailability of granular data among the different PLVAD devices used, we were unable to perform separate subgroup analysis for Impella and tandem heart as the studies that include both devices do not clearly separate the the events among the devices but have pooled data. As per the request of the reviewer, we have explored further on this in detail under limitations.
Page 14 under 4.2 Strength & Limitations section.
We have added the findings from the recent PRAISE registry data which shows outcomes between STEMI and NSTEMI and how it affects device selection in detail in the discussion as suggested by the reviewer
Page 15, under 4.3 Future directions section.
Comment 4: Finally, some forest plots are suboptimally formatted, lacking detailed axis labels, consistent confidence interval shading, and clear risk tables. The authors should improve this aspect by using uniform labeling, adding 95% confidence interval bars, and including risk tables beneath the plots.
Response 4:Thank you for pointing this out. We have optimally formatted all our forest plots with uniform labeling and 95% confidence interval bars, and the ROB table is separately listed.
Page 8, 9, 10, 11 under 3. Results section.

Round 2
Reviewer 2 Report
Comments and Suggestions for Authors
My congratulations to the authors for their effort in revising the manuscript.
Author Response
Comment 1: My congratulations to the authors for their effort in revising the manuscript.
Response 1: thank you for reviewing our manuscript and for the insightful comments